# Ellagic Acid: A Green Multi-Target Weapon That Reduces Oxidative Stress and Inflammation to Prevent and Improve the Condition of Alzheimer’s Disease

**DOI:** 10.3390/ijms26020844

**Published:** 2025-01-20

**Authors:** Silvana Alfei, Guendalina Zuccari

**Affiliations:** 1Department of Pharmacy (DIFAR), University of Genoa, Viale Cembrano, 4, 16148 Genova, Italy; 2Laboratory of Experimental Therapies in Oncology, IRCCS Istituto Giannina Gaslini, Via G. Gaslini 5, 16147 Genoa, Italy

**Keywords:** Alzheimer’s disease (AD), one-target drugs, multi-target drugs, oxidative stress (OS), reactive oxygen species, reactive nitrogen species, antioxidants, radical scavenging activity, ellagitannins (ETs), ellagic acid (EA), urolithins (UROs), in vitro and in vivo EA applications, AD diagnosis

## Abstract

Oxidative stress (OS), generated by the overrun of reactive species of oxygen and nitrogen (RONS), is the key cause of several human diseases. With inflammation, OS is responsible for the onset and development of clinical signs and the pathological hallmarks of Alzheimer’s disease (AD). AD is a multifactorial chronic neurodegenerative syndrome indicated by a form of progressive dementia associated with aging. While one-target drugs only soften its symptoms while generating drug resistance, multi-target polyphenols from fruits and vegetables, such as ellagitannins (ETs), ellagic acid (EA), and urolithins (UROs), having potent antioxidant and radical scavenging effects capable of counteracting OS, could be new green options to treat human degenerative diseases, thus representing hopeful alternatives and/or adjuvants to one-target drugs to ameliorate AD. Unfortunately, in vivo ETs are not absorbed, while providing mainly ellagic acid (EA), which, due to its trivial water-solubility and first-pass effect, metabolizes in the intestine to yield UROs, or irreversible binding to cellular DNA and proteins, which have very low bioavailability, thus failing as a therapeutic in vivo. Currently, only UROs have confirmed the beneficial effect demonstrated in vitro by reaching tissues to the extent necessary for therapeutic outcomes. Unfortunately, upon the administration of food rich in ETs or ETs and EA, URO formation is affected by extreme interindividual variability that renders them unreliable as novel clinically usable drugs. Significant attention has therefore been paid specifically to multitarget EA, which is incessantly investigated as such or nanotechnologically manipulated to be a potential “lead compound” with protective action toward AD. An overview of the multi-factorial and multi-target aspects that characterize AD and polyphenol activity, respectively, as well as the traditional and/or innovative clinical treatments available to treat AD, constitutes the opening of this work. Upon focus on the pathophysiology of OS and on EA’s chemical features and mechanisms leading to its antioxidant activity, an all-around updated analysis of the current EA-rich foods and EA involvement in the field of AD is provided. The possible clinical usage of EA to treat AD is discussed, reporting results of its applications in vitro, in vivo, and during clinical trials. A critical view of the need for more extensive use of the most rapid diagnostic methods to detect AD from its early symptoms is also included in this work.

## 1. Introduction

### 1.1. Methods

To select the literature material useful to edit this review, we used keywords like those reported in the keywords section. Specifically, in Scopus, PubMed, and PubChem databases, we used the following keywords: Alzheimer’s disease; one-target drugs, multi-target drugs; oxidative stress; reactive oxygen and nitrogen species; antioxidant effects; radical scavenging activity; ellagitannins; ellagic acid; urolithins; in vitro and in vivo ellagic acid applications; and Alzheimer’s disease early diagnosis. Using this method, several reviews and experimental works were collected, whose number was immediately reduced by removing duplicates. The remaining works were further reduced to the 310 references used in this study by removing the obsolete ones and then by dividing the residuals into experimental and review works. The first ones were used to obtain the most updated information and recent advances concerning Alzheimer’s disease (AD), the current drugs available to treat its symptoms, and the possible new curative options under investigation, including polyphenols. The review articles were also used to obtain information on the advantages and disadvantages of ellagitannins, ellagic acid, and urolithins in terms of biomedical applications, their main sources, and the mechanisms of their antioxidant powers as the basis of their beneficial properties. On the contrary, the experimental studies provided us with data useful to organize the tables containing information about the most relevant findings achieved with the in vivo and in vitro applications of urolithin and ellagic acid (EA).

### 1.2. Background

#### 1.2.1. Alzheimer’s Disease

Alzheimer–Perusini disease, mainly known as Alzheimer’s disease (AD), presenile dementia of the Alzheimer’s type, primary degenerative dementia of the Alzheimer’s type, and, for simplicity, Alzheimer’s, is the most common form of progressively disabling degenerative dementia, with onset mainly in presenile age, specifically over 65 years [1]. It is estimated that approximately 50–70% of cases of dementia are due to the AD condition, while 10–20% are due to vascular dementia [2]. Some data from the World Alzheimer Report 2023 produced by Alzheimer’s Disease International established that in the next 25 years, the number of people living with dementia worldwide could increase from 55 million to 139 million. Furthermore, the costs associated with the disease could jump from USD 1.3 trillion in 2019 to over 2.8 in 2030. The most frequent early symptom is represented by difficulty in remembering recent events, followed by other symptoms that may appear with aging, including aphasia, disorientation, sudden changes in mood, depression, inability to take care of oneself, and behavioural problems. Also, confusion, irritability and aggressiveness, mood swings, difficulty speaking, both short- and long-term memory loss, and progressive sensory dysfunction further aggravate the already detrimental condition of patients suffering from AD [3,4]. The subject tends to isolate himself from society and family, and gradually, basic mental abilities are lost. It seems that about 70% of AD development is genetic, with several genes usually involved. However, the exact cause and progression of AD are still not well understood. It is well established that AD is a well-unshakable neuronal dysfunction whose primary causes could be associated with toxin insults, heredity, metabolism, or even attack by infectious pathogens [5]. Several research studies indicate that AD is closely correlated with amyloid plaques and neurofibrillary tangles found in the brain, but the root cause of this degeneration is unknown [6]. Other well-explored factors contributing to cognitive neurodegeneration driving AD comprise excessive acetylcholine esterase enzymes (AChE), β amyloid (βA) precursor protein-cleaving enzyme 1 (BACE-1), glycogen synthase kinase 3 β (GSK-3 β), monoamine oxidases (MAOs), metal ions in the brain, N-methyl-D-aspartate (NMDA) receptor, and phosphodiesterase (PDE). It is extensively recognized that OS, as well as the formation of free radicals and not radical RONS, are strongly involved in the progression of brain aging and in the onset and evolution of AD. In addition, impaired bioenergetics, mitochondrial abnormalities, and neuroinflammatory processes are implicated. Collectively, one hundred years after AD discovery, experts in the field are confident in asserting that, even if AD’s pathogenesis is not yet entirely understood, it is a multifactorial disorder whose causes can be genetic, environmental, and endogenous (Figure 1), like other neurodegenerative diseases [7]. The excessive incorrect folding and aggregation of proteins often related to the ubiquitin–proteasomal system (UPS) are also accountable to AD.

Particularly, the increase in RONS causes mitochondria and DNA damage, with increased production of toxic Aβ causing, in turn, severe DNA repair dysfunctions. Currently, approved therapeutic treatments used to treat AD provide only little and temporary benefits to symptoms and can partially slow the progression of the disease. Increasing insights, coupled with further ongoing discoveries about AD multi-factorial pathogenesis, have provided the rationale for the search for new therapies that could directly target the molecular causes of AD [7]. New drug candidates with promising potential to modify the disease are now under clinical trials [9]. On 1 January 2023, 141 exclusive treatments for AD were under investigation in 187 trials. Specifically, 36 agents were in 55 Phase 3 trials, 87 agents were in 99 Phase 2 ones, and 31 agents were in 33 Phase 1 ones. Among these, 79% of drugs in trials were those proposed as disease-modifying therapies, while 28% of therapies under experimentation were those using repurposed agents. Jointly, existing Phase 1, 2, and 3 trials have a need for 57,465 participants [9]. Unfortunately, although over 500 clinical trials have been conducted to identify a possible effective treatment for AD, no treatment has yet been identified capable of halting or reversing the disease [10]. The widespread and increasing diffusion of AD in the population and the limited and non-resolving efficacy of the available therapies, as well as the enormous resources necessary for its diagnosis and management in terms of social, emotional, organizational, and economic aspects, make AD one of the diseases with the most serious social impact in the world [11]. This lack of pathogenesis-targeting therapies is principally due to the limiting effects of the blood–brain barrier (BBB), which keeps out of the brain about 99% of all “foreign substances”. After their discovery, nanoparticles (NPs) have been successfully used for targeted delivery into many organs, including the brain [12]. In this context, new nano-dimensional agents and/or formulations of existing drugs could be promising options for the possible diagnosis and treatment of various neurological disorders, including AD. Furthermore, it has been reported that drugs striking a single molecular target are not suitable to treat disorders like neurodegenerative and cardiovascular ones, diabetes, cancer, etc., which embrace multiple factors of pathogenesis [13]. On the contrary, drugs that involve different pharmacological approaches could provide more potential methods of overcoming the obstacles that could occur upon the use of single-target drugs, often well-functioning in vitro but not in vivo experiments.

#### 1.2.2. Medical Potentialities of Fruits and Vegetables

In this worrying scenario concerning the poor available arsenal of drugs and/or nano-drugs to treat AD, the several multitarget health effects of many fruits and vegetables could represent an appealing alternative treatment option. In fact, it has been demonstrated that foods including muscadine grape; berries such as pomegranates, strawberries, raspberries, and blackberries; nuts such as chestnuts, walnuts, almonds, pecans, and pistachios; herbs such as *Camellia sinensis*, seeds including berry seeds; and their derived foods and/or beverages possess recognized healthy and/or preventive effects against several complex human diseases, thus evidencing their multitarget behavior [14]. Such effects have been mostly associated with their high content of antioxidant molecules, mainly polyphenols [14,15,16], such ellagitannins (ETs), as well as gallic acid (GA) and ellagic acid (EA), which are produced via their hydrolysis in vivo (Figure 2) [17]. By limiting the hyperproduction of RONS, they counteract OS, recognized as the foremost prompting factor of several human discomforts.

Particularly, the strong correlation existing between the intake of foods containing high amounts of ETs and the subsequent beneficial effect vs. several human degenerative diseases is extensively reported [17,18]. As examples, documented findings assert the existence of an association between the eating of foods rich in ETs and stronger cardiovascular health [19,20], or between the intake of fruits and vegetables and minor occurrence of coronary heart disease [21]. Much experimental data led to the assumption that ETs might be used to prevent difficult-to-treat disorders such as those of diabetes, cancer, cardiovascular diseases, and the central nervous system (CNS), including AD [22]. Nonetheless, in Europe, the European Food Safety Agency has not still approved any kind of health claims concerning ETs [14]. As mentioned above, ETs are capable of providing EA via hydrolysis, which is rationally considered the bioactive fragment of ETs. Indeed, possessing one of the strongest antioxidant powers, it is the molecule capable of counteracting OS to an extent that can be helpful in preventing and/or ameliorating AD conditions [17], as confirmed 10 years ago in a study by Kilic [23]. The in vitro radical scavenging and antioxidant capacity of EA was clarified using different analytical methodologies such as total antioxidant activity determination via ferric thiocyanate, hydrogen peroxide scavenging, 1,1-diphenyl-2-picryl-hydrazyl free radical (DPPH) scavenging, 2,2′-azino-bis(3-ethylbenzthiazoline-6-sulfonic acid) (ABTS) radical scavenging activity and superoxide anion radical scavenging, ferrous ion (Fe^2+^) chelating activity, and ferric ion (Fe^3+^) reducing ability [23]. Being endowed with this relevant capability to combat OS, nowadays considered the key cause of all diseases, and therefore being gifted with the capacity to ameliorate human degenerative diseases, food chemists consider both ETs and EA as nutraceuticals (NTs). NTs are defined as compounds that possess both canonical nutritional values and several additional health benefits. In this regard, a diet of NTs-rich foods often triggers relevant positive biological effects. Anyway, despite this definition, as mentioned above and more deeply discussed later, ETs are not absorbed in vivo and are not capable of reaching tissues and organs, thus being unsuitable as molecules or template molecules to develop new treatments for human diseases. Specifically, they are hydrolysed in the gastrointestinal tract (GIT), thus providing, among other molecules, EA, which, possessing the strongest antioxidant effects and beneficial properties, represents an excellent platform to develop new drugs for AD. Unfortunately, EA also undergoes massive metabolism to urolithins (UROs) in the intestine, which are still endowed with appreciable beneficial properties, but studies have demonstrated that UROs are not advisable for harmless therapeutic purposes due to their double-faced behaviour. They can induce beneficial effects, but on the basis of their chemical structure, environmental settings, the sort of target cells under study, the individual’s age, and their health state, they could also lead to harmful consequences [17]. In this scenario, EA remains the polyphenolic small molecule that attracts the interest of researchers as a promising molecule to provide benefits in neurodegenerative disorders, including AD. In this context, the main challenges of researchers in the field include defining which pharmacophore/pharmacophores in EA can be actually responsible/s either for its health benefits or for its possible collateral effects. Also, the incessant development of new EA dosage forms capable of improving its bioavailability and in silico screening investigations to design new multi-target EA-type CNS drugs are the goals of chemists, pharmacologists, and practitioners.

### 1.3. Our Aims

Given the above-reported considerations, this review aimed at more largely driving researchers’ attention toward EA as an actual possible multi-target treatment option for AD.

The main purpose of this manuscript is to stimulate major interest in EA and to increase research on it as a promising platform to develop more effective compounds for AD, including EA nano-formulation, to solve its bioavailability drawbacks. Solving important issues or filling gaps in this field was not in the scope of this review, whose main goal was to provide all-round knowledge about AD as a multifactorial neurodegenerative disease and its supposed causes, as well as polyphenols and mainly EA, in addition to their antioxidant mechanisms responsible for their beneficial effects on AD.

### 1.4. Summary

A brief overview of the multi-factorial and multi-target aspects that characterize AD, as well as polyphenols such as EA, respectively, open this work. Focusing on the pathophysiology of OS, EA chemical features, and the mechanisms of its antioxidant activity, an all-around updated analysis concerning EA-rich foods and EA involvement in the field of AD is provided. The possible clinical usage of EA to treat AD is shown, reporting results by its applications in vivo and clinical trials. A critical view of the need for more extensive use of the most rapid diagnostic methods to detect AD from its early symptoms is also included in this work.

## 2. Multifactorial Nature of Neurodegenerative Diseases: Alzheimer’s Disease (AD)

For years, neurodegenerative disorders (NDs) have been considered the most mysterious and challenging problems in medicine and biomedicine [12]. While moving from descriptive phenomenology to mechanistic analysis, researchers have become progressively aware that the major processes involved in their onset are complex and multifactorial, including genetic, environmental, and endogenous factors [24,25]. Such NDs comprehend, among others, amyotrophic lateral sclerosis (ALS), multiple sclerosis (MS), Parkinson’s disease (PD), Alzheimer’s disease (AD), Huntington’s disease (HD), multiple system atrophy (MSA), tauopathies (TPs), and prion diseases PRD). As in other neurodegenerative conditions, the pathogenic cascade driving AD includes protein non-correct folding and combination, RONS non-controlled production, and, consequently, the onset of OS, metal dyshomeostasis, mitochondrial impairments, and phosphorylation dysfunctions, all occurring concomitantly. Figure 3 summarizes the concomitant multiple factors leading to the onset of AD conditions, while Figure 4 evidences how some of these factors can directly damage neurons, causing their death or potentially triggering a detrimental cascade of events anyway, leading to the death of neurons.

Protein misfolding, followed by self-association and the subsequent deposition of aggregation supported by OS, increase in uncontrolled RONS, and metal dyshomeostasis, has been observed in the brain tissues of patients affected by AD [26]. The findings suggest that protein assemblies produced by different amyloidogenic proteins share common structural and histological morphologies and might trigger similar neurotoxic mechanisms. The biophysical behavior of these proteins, leading to their incorrect folding, aggregation, and deposition, has prompted scientists to group these kinds of neurological disorders under the common name of “conformational diseases” [27]. It is worth noting that amyloid oligomers such as amyloid-precursor protein (A) and R-synuclein have been widely reported to permeabilize both cell and mitochondrial membranes, thus impairing their functions [28]. They are, therefore, probably responsible for subsequent abnormal calcium regulation, depolarization of membranes, and reduced mitochondrial functions, which have been commonly detected in AD conditions [29].

### More in Deep in the Multifactorial Causes of AD: Reactive Oxygen and Nitrogen Species (RONS)

The role of RONS in many NDs is essential. While the physiological production of RONS generation is controlled well by the antioxidant fortification and repair systems of cells [30], when overproduced, they disrupt the cell’s detoxification systems, which fail to maintain RONS levels within the correct parameters. As a result, they could reach high concentrations, thus causing OS and inflammation. Irremediable damage to nucleic acids, lipids, and proteins happens, thus encouraging aging, age-related disorders, and most degenerative human diseases [30]. To respond to the answer “Is OS a cause or a consequence of the neurodegenerative cascade in AD?” has been and remains a daily challenge for experts in the field, which urgently requires a solution. At present, scientists agree almost unanimously to affirm that the unbalanced intracellular state of oxidation is the first event in neurodegeneration and is one of the most important factors in neurodegenerative disorders. Neurons are particularly vulnerable to OS, and the possible inequity in pro-oxidant/antioxidant homeostasis in the central nervous system (CNS) can translate into the further production of several potentially toxic RONS, including both radical and nonradical species, that contribute to the onset and/or propagation of radical chain reactions damaging neurons. Table 1 reports the possible sources of RONS, which can be endogenous, both enzymatic and non-enzymatic, as well as exogenous.

Figure 5 specifically schematizes the main endogenous processes by which ROS can be created in cells and the detrimental effects they can have on health [30], including DNA damage, lipid and protein peroxidation, telomere reduction, aging, and death.

In AD, OS has been detected in every family of molecules within neurons, including lipids, DNA, and proteins. Several clinical studies have revealed that the simple administration of one or a few one-target antioxidants had modest success in the treatment of neurodegeneration. It has been reported that in AD, a direct cause/effect relationship between metal impairments and heightened oxidative damage exists. Transition metals such as iron, copper, or other redox active metals are pivotal in several biological reactions, but their homeostasis alteration may drive increased free radical production. Moreover, while all the disease-specific proteins exhibit metal-binding items, metal ions promote the generation of fibrils, and the deposition of proteins found in AD (Section 2, Figure 4). Furthermore, metal-mediated OS is not only a cause of OS in neurons but also of mitochondrial impairments, where RONS can also be produced. Morphological, biochemical, and molecular irregularities in mitochondria present in different tissues of AD patients have been signalled. Although the chronological hierarchy of events and underlying causes in AD concerning mitochondrial dysfunction and OS are not yet fully elucidated, there are unequivocable marks that both support the development of the others, actuating a self-sustaining, intensifying cycle that ultimately triggers neuronal death processes, as shown in Figure 6.

Also, the endoplasmic reticulum (ER) is an essential apoptotic factor. It has been shown that, in AD, apoptosis provoked by badly bent proteins encompasses ER impairment. Moreover, the alteration of the state of phosphorylation of some pivotal proteins involved in the pathogenic cascades represents an additional mechanism usually shared by NDs. In addition to the extensively recognized hyperphosphorylated state of τ protein in the neurofibrillary tangles observed in AD brain, other specific impaired kinase and phosphatase activities are coupled to alterations in the phosphorylation state of disease-specific proteins, which are specific for PD, ALS, and HD. Several molecular evidence demonstrated a cell-type specificity in neuronal disorders and selective neuron degeneration in AD. Nevertheless, these general mechanisms alone are not sufficient to explain the high number of biochemical and pathological abnormalities of AD. Collectively, disfunctions in AD incorporate a multitude of cross-related cellular and biochemical changes that cannot be effectively addressed by using treatments based on a one-molecule, one-target paradigm. In our opinion, there is a growing interest and urgent need for the development of multi-target directed ligands (MTDLs) to provide real disease-modifying drug candidates for this ND.

## 3. One-Target Drugs vs. Multi-Target Therapies in the Treatment of Degenerative Diseases

Scientific knowledge about the pathogenesis of several human diseases has advanced enormously in recent decades. Therefore, the sector of drug discovery has gradually moved from pursuing a completely human phenotype-based tactic to a simpler approach based on single molecular targets. This revolution has led to a type of drug research still extensively followed, which is focused on the discovery of small molecules capable of regulating the biological function of a single target alleged to be fully responsible for a given disease. The research in this direction has been finalized to discover drug molecules selective for a given protein. Nowadays, many ligands endowed with outstanding in vitro selectivity and efficacy are available. Unfortunately, it should be noted that a highly selective ligand for a certain target in vitro is not always also a clinically efficacious drug in vivo (Table 2).

The low correspondence between the results in vitro and those in vivo in the case of NDs is mainly due to the multifactorial nature of human degenerative diseases. In NDs, the cells often find strategies to make up for a single protein, whose activity is influenced by the one-target drug administered due to the redundancy of the system, including the existence of parallel pathways [31]. Drugs striking a single target are commonly inadequate to treat diseases like neurodegenerative ones, including AD, diabetes, cardiovascular disorders, and cancer, which involve multiple pathogenic factors [32]. Different pharmacological strategies are necessary to overcome the problems that arise from the use of single-target drugs (Table 2, column 1). When a single-target drug is not appropriate to successfully cure a disease, alternative options aiming at hitting more than one impaired process correlated to the disorder should be considered. Figure A1 in Appendix A at the end of this manuscript shows some alternative medical approaches.

The three most commonly adopted approaches (MMT, MCM, and MTDLs) reported in Figure A1 are charted in Table 3 with related advantages and disadvantages.

The multiple-medication therapy (MMT) (Figure A1), also known as combination therapy, may be used as an alternative option to one-target therapy. It is usually composed of two or three different drugs singularly administrated, thus combining different therapeutic mechanisms [36]. A second tactic could concern the use of a multiple-compound medication (MCM), also referred to as a “single-pill drug combination”, which involves the inclusion of different drugs into the same dosage form. Finally, a very appealing strategy is now appearing, which assumes that a single compound may be able to hit multiple targets per se because it comprehends in the same molecule more than one pharmacophore. Obviously, there are a series of multiple advantages over MMT or MCM inherent to possible therapies using a single multitarget drug (Table 3). Indeed, there is a solid suggestion that the development of single compounds able to strike multiple targets might reveal new opportunities for the treatment of major NDs, such as AD, for which new effective cures are an urgent need and an unmet goal. In the past, Morphy and Rankovic pleasingly discussed this approach in three articles, which were mostly concerned with non-NDs [37,38,39]. In this context, we are convinced that the definition “multi-target-directed ligands” (MTDLs) more completely describes these compounds. Effectively, MTDLs should succeed in treating complex diseases because of their ability to interact with the multiple targets thought to be responsible for disease pathogenesis. The excellent perspective by Morphy and Rankovic [37] covered several aspects of the design strategy leading to MTDLs for different areas such as inflammation, dopaminergic D2-receptors, histaminergic H1-receptors, serotoninergic receptors, angiotensin system, peroxisome proliferators activated receptors, kinases, and nitric oxide-releasing conjugates. Although more attention to the achievements of MTDLs for NDs is increasing, there is still a paucity of review literature dealing with complex diseases associated with neurodegeneration, which we hope to compensate for in our present work.

### 3.1. Alzheimer’s Disease (AD) and Currently Available Medicines and/or Treatments in Development

Among the NDs reported above, AD stands out as the fourth leading cause of death in Western countries and the most common cause of acquired dementia in the elderly population. As shown in Figure A2 (Appendix A), two main forms of AD are recognized, both characterized by neuronal death.

In line with an increase in the average life expectancy of humans, the number of affected persons is expected to triple by 2050, with immense economic and personal tolls [35]. In parallel with this increase, the speed of drug research has accelerated noticeably in recent decades, but not enough. However, the number of therapeutic options on the market remains strongly restricted. Worryingly, the currently registered drugs for AD, i.e., acetylcholinesterase inhibitors (AChEIs), are not able to alter or prevent disease progression. They are instead palliative in alleviating disease symptomatology [40]. In this scenario, where AD is a multifactorial disease and insights and discoveries about its pathogenesis are progressively ongoing, the rationale exists for the discovery and study of multi-target drugs directly targeting different AD molecular causes simultaneously.

#### 3.1.1. Current AD Therapies

Although the path of the events leading to AD onset is far from being completely clarified, the cholinergic hypothesis is the oldest one and had the strongest influence on the development of clinical treatment strategies for AD. Acetylcholine (Ach) is released in the synaptic cleft, where it activates both postsynaptic and presynaptic cholinergic receptors [nicotinic (N) and muscarinic (M)], leading to an increase in cholinergic transmission, which results in cognition improvement. Anyway, ACh is removed from the synapse by the action of the enzyme acetylcholinesterase (AChE), which, therefore, has become the target for the development and approval of acetylcholinesterase inhibitors (AChEIs) for AD treatment, as visualized in [Fig ijms-26-00844-ch001] and reported in Table 4.

The acetylcholinesterase inhibitor (AChEI) tacrine ([Fig ijms-26-00844-ch001]) was the first drug to be approved for the treatment of AD but is now rarely used because of its hepatotoxicity. Later, three other AChEIs, donepezil, rivastigmine, and galantamine, reached the market, becoming the standard for AD therapy, only later complemented by memantine, a noncompetitive NMDA antagonist ([Fig ijms-26-00844-ch001]). Table 4 includes the advantages and disadvantages connected to the use of such therapeutics.

Despite the diffused clinical practice, the debate on the effective activity of AChEI medications endures. So, the search for novel AChEIs, such as inhibitors of the “non-classical function” of AChE, has rehabilitated interest in expanding their potential as real disease-modifying agents. Current AD drug development programs focus primarily on agents with anti-amyloid disease-modifying properties, and several studies have been carried out on molecules capable of reducing amyloid pathology (Table 5). Classes of therapeutic modalities currently in the advanced stage of clinical trial testing comprise forms of immunotherapy that use several drugs (Table 5), including medicaments with anti-amyloid properties. Nontraditional dementia therapies, such as those using HMG-CoA reductase inhibitors, mainly including statins [42], such as atorvastatin, simvastatin, fluvastatin, pravastatin, rosuvastatin, and lovastatin, are now being evaluated for their clinical benefits in AD as disease-modifying treatments [42].

#### 3.1.2. Versus Disease-Modifying Therapies in Alzheimer’s Disease [123]

The long-expected era of disease-modifying therapy (DMT) for AD has finally arrived and will substantially influence how the disease is perceived and managed. Unfortunately, the new treatments closest to extensive clinical implementation (Figure A3, Appendix A) will pose challenges for rightful access. No national healthcare system is ready to deliver these drugs to more than a fraction of patients who might be eligible.

These active principles (APs) include lecanemab and donanemab, which are intravenous monoclonal antibodies capable of removing βA plaques from the brain, thus slowing cognitive and functional decline. Paradoxically, lecanemab and donanemab have revealed side effects, mainly amyloid-related imaging abnormalities (ARIA), in about 21% and 39% of patients, respectively [124]. While usually asymptomatic and transient, ARIA requires close monitoring. Symptoms and signs of ARIA can be non-specific, including blurred vision, headaches, and unsteadiness, or can include focal deficits such as dysphasia. However, many patients with ARIA can be re-dosed safely after a period of treatment [124].

#### 3.1.3. Multi-Target Therapy (MTT) for AD

However, the adoption of MMT, MCM, and MDTLs (or MTSM) might result in more effective treatment strategies for AD due to the multifactorial nature of this disorder. MMT has already proven successful in the treatment of other complex diseases such as cancer, HIV, and hypertension. Due to the possibility of attacking several targets simultaneously, exploiting synergy, and minimizing the individual toxicity of the administered single drugs, maximum efficacy has been achieved. With similar outcomes and advantages, MCMs were used to ameliorate the compliance of patients with AD. Since 2006, the number of patented MCMs, where new compounds that revealed potentialities to ameliorate AD were administered in combination with old therapeutics (AChEIs or NDMA receptors antagonists, as well as NSAID or a combination of two), has overtaken that of single-drug entities for the potential treatment of AD [125] (Table 6).

In the clinic, the MMT of memantine *plus* an AChEI appears to produce an additional effect, resulting in a well-tolerated, effective treatment strategy [137]. Considering the well-accepted clinical use of MMT only as a starting point, the MTDL design strategy might represent its natural evolution, and MTDLs emerge as valuable tools for better hitting the multiple targets implicated in AD etiology [138]. Several MTDLs have been developed by academia and industry in recent years. These have been the subject of some interesting review articles, and particularly interested readers could examine the related references [139,140,141,142]. The main design strategy usually applied to build up a possible new MTDL involves detecting the active portions of different drugs and combining them in a single structure to afford hybrid molecules [8]. In principle, each pharmacophore of these new drugs should retain the ability to interact with its specific site(s) on the target and, consequently, produce specific pharmacological responses that, taken together, should slow or block the neurodegenerative process of AD. Specifically, it is in use to modify the molecular structure of an AChEI by inserting opportune pharmacophores (indicated as PG groups in Figure 7) already present inside other drugs, which demonstrated beneficial effects in neurodegenerative diseases, to provide the traditional drug with additional ameliorative effects while reducing the side effects of separate single drugs and enhancing the compliance of patients [8].

## 4. Ellagitannins (ETs) and EA as Multi-Target Compounds: Strengths and Weaknesses

Both ETs and EA have proven, at least in vitro, to prevent and/or ameliorate chronic diseases such as cancer, diabetes, and those of the cardiovascular system [143], and, lately, neurodegenerative diseases [144,145]. It seems that these positive effects are due to their multi-target action accounting for anti-angiogenic, anti-atherogenic, anti-carcinogenic, anti-obesity, anti-inflammatory, antioxidant, and anti-thrombotic properties, together with anti-neurodegenerative capability. All these gains seem to derive from their antioxidant power and, therefore, their capability to contain OS, the key cause of all human disorders [14,17]. Since neurodegenerative disorders, including AD, are multifactorial diseases, the application of the usual and extensively approached one-molecule, one-target paradigm, providing drugs able to hit only a single target, could have limited effects, mainly in vivo, and may also translate in the emergence of resistance. On the contrary, a compound capable of interfering with different targets involved in the cascade of pathological events leading to a given disease could be highly effective for treating multifactorial diseases, such as AD [13]. The synthetic design of such drugs may not be easy, because the obtained drugs could bind in vivo targets that are not involved with the disease of interest and could not necessarily be responsible for side effects. On the contrary, natural polyphenols such as ETs and EA, per se possessing the multifaceted health activity reported above as demonstrated by the outcomes deriving by the assumption of food containing them, are provided readily by nature and could be promising options to ameliorate/treat AD. However, they could serve at least as template molecules to be used as starting platforms to design new multi-target drugs.

### 4.1. Bioavailability Drawbacks Associated with ETs and EA

According to a review reported in 2020, except for an insignificant quantity (e.g., 0.7–4.7 mg/100 g of berries, wet weight), the free form of EA mainly has its origin in vivo, after the consumption of ETs-rich food, due to the physiological massive hydrolysis of ETs in the gastrointestinal tract (GIT) [17]. Anyway, even if, according to some other authors, free EA makes up only a small part of the total EA pool in plants, others suggest that its portion can reach and even exceed 50% of the total content, depending on the plant species. Interestingly, in the fruits of *Terminalia ferdinandiana* Exell, a native Australian plant known as the Kakadu plum, EA was found to be mostly free form, with a percentage reaching 70.6% of the total EA pool [146]. By contrast, the percentage of free EA in strawberries, as shown by the same study, reached only 7.4% of its total content [146]. Despite early studies not showing the presence of EA in plants of the Fabaceae family, there is now evidence of relatively high levels of this phytochemical in several sprouted legumes, such as sprouted adzuki bean (*Vigna angularis*), some varieties of bean (*Phaseolus vulgaris* L.), cowpea (*Vigna unguiculata* (L.) Walp.), pea (*Pisum sativum* L.), and soybean *(Glycine max* (L.) Merr.) [147]. Sprouted soybeans have been found to have a considerably higher EA content than other sprouted legumes (45.6–48.9 mg/100 g vs. 8.96–18.3 mg/100 g dry weight) [147]. Although the ratio between free and bound forms of EA in plants may vary considerably depending on the plant species, the proportion of unbound EA may also depend on the method chosen for determination, the type of storage, and the processing practice [148]. Freezing fruits, as well as processing them to produce beverages and jams, may have different effects on the content of EA. However, after the intake of ETs-rich foods, ETs are only slightly absorbed and reach the small intestine, where they are hydrolyzed to EA by the gut’s microbiota action [17]. Once produced, EA is practically not absorbed due to its trivial water solubility, unfavorable physicochemical characteristics, and low bioavailability (Table A1, Appendix A) and reaches the large intestine untouched. A justification for EA’s poor bioavailability and its low concentrations in plasma and tissues depends mainly on its tendency to tie up permanently cellular nucleic acids and proteins or to form weakly solvable aggregates with the ionic form of calcium and magnesium, which greatly reduce transcellular absorption [149]. Also, still-active metabolites of EA were sparsely detected in fluids at 1 and 5 h after ingestion, thus corresponding to very low concentrations as well, not enough to supply substantial positive effects [17]. In the large intestine, EA is metabolized to the more hydrophilic urolithins (UROs), secondary polyphenol metabolites derived from the gut microbial action [150], and/or converted to its dimethyl, as well as dimethyl glucuronate and sulphate derivatives, which are excreted.

A representative structure of an ET (casuarictin); that of EA; and those of URO A, B, C, D, iso-A, and iso-B are shown in Figure 1, which shows the path of EA formation after the intake of ETs-rich foods and its subsequent metabolism to UROs and dimethyl ether derivatives [17]. A more recent article has also introduced URO-M5 and M6 among the URO-type metabolites of EA [150]. Precisely, in this new route, EA is transformed into URO-M5, which is in turn converted into URO-D, while URO-M5 is converted into URO-M6, which then provides URO-C as URO-D [150].

In the year 2022, a study reported the existence and structure of up to 13 UROs [151]. Collectively, since ETs are poorly adsorbed in GIT, they cannot reach blood and tissues, where they could exert their beneficial effects but provide the bioactive EA upon hydrolysis. Nonetheless, instead of being absorbed and reaching blood and tissues, due to its very low water solubility [152], EA also undergoes a massive metabolism. Specifically, it is transformed in UROs and in other metabolites excretable with urine, and the amount of EA detected in blood and tissues observed after ETs-rich foods intake is insignificant in improving the conditions associated with chronic human diseases. Due to this process, the findings obtained with ETs and EA in vivo studies against several human pathologies did not coincide with the promising ones observed in vitro, as generally happens for dietary polyphenols [14,153]. As observable in Figure 1, UROs are dibenzopyran-6-one derivatives with different hydroxyl substitutions. UROs are more lipophilic than EA, and this characteristic is responsible for their greater absorption rate respect to EA, thus being the only active phenolic molecules sufficiently absorbed and detectable in the circle and cells after ETs-rich food intake [150]. URO-A and URO-B are the major metabolites of EA found in the gut, where URO-A is the most biologically active compared to the rest of the EA metabolites [150]. In enterocytes and hepatocytes, UROs undergo biotransformation to UROs metabolites. UROs’ main metabolites detected in plasma and urine consist of their glucuronic and sulfate conjugates, such as URO-A and URO-B glucuronide and sulfate, while the minor metabolites are URO-C and iso-URO-A glucuronide.

### 4.2. Ellagic Acid or Urolithins?

Apparently promising, in vitro and in vivo experiments have also revealed that UROs have anti-inflammatory, anti-carcinogenic, anti-glycative, antioxidant (even if lower than ETs and EA), and antimicrobial properties. They exert preventive effects on gut and systemic inflammation and also seem to play the role of hormone analogues [154]. Table 7 reports the most relevant studies concerning the in vivo effects of UROs assessed in animal models.

Due to the confirmations both in vitro and in vivo about the pharmacological properties of UROs, currently, there is an extensive tendency to think that UROs, rather than EA, could be the effective bioactive molecules accountable for the beneficial outcomes deriving from the intake of foods rich in ETs and EA [14,67]. This proposition is assisted by the awareness that, although in vitro findings have demonstrated that EA and UROs are almost equally active, in vivo studies only provided trustworthy verification about this fact with regard to UROs. Only UROs have been found in fluids, cells, and tissues and were measured, finding concentrations capable of exerting the ameliorative effects already evidenced in vitro. On the other hand, the poor in vivo reliability of UROS (see the next paragraph for details), the greatest antioxidant effects peculiar to EA, which could be of greater help in ameliorating neurodegenerative disorders, including AD, have stimulated the interest of researchers in knowing more about the possible EA activity in vivo if absorbed. This has led scientists to increasingly and incessantly focus on preparing water soluble and absorbable EA formulations able to defend EA and to lower or annul EA metabolism to UROs so that it could reach cells and tissue in pristine form [188]. The formulation of drug delivery systems capable of transporting and releasing EA to the target site represents a valid approach for bypassing the bad biopharmaceutical features of this polyphenol, thus allowing a better evaluation of its potential application as a radical scavenger antioxidant therapeutic. In this context, after the year 2019, we studied some micro- and nanosized solutions, which revealed interesting performance [189,190,191].

### 4.3. Drawbacks Associated with UROs Hamper Their Clinical Development, Thus Quenching Researchers’ Interest

Although gifted with beneficial characteristics like those of EA, UROs are not appropriate for secure therapeutic use due to their double-faced behaviour. Depending on their chemical structure, environmental settings, the class of target cells studied, individual age, and their health state, they could also be dangerous [17]. The amount and typology of UROs produced in the gut of individuals also depend on the type of vegetables that have been introduced and the individual microbiota metabolic activity—that is, typified by a highly inter-individual heterogeneity, depending on several factors and human metabotypes (0, A, and B) [17]. Moreover, this highly interindividual and intra-individual process is not completely elucidated yet [34,35]. Let us imagine that even living species that do not produce UROs exist. Table 8 reports the UROs mainly found in different mammalian species after the consumption of different vegetables.

URO absorption, blood and tissue concentrations, and inter-subject variability in the comebacks to URO exposure are arbitrary variables that drive various responses that, ironically, could promote adverse effects. In addition, human microbiota activity is difficult to reproduce in animal models and cannot be easily studied and/or controlled [17].

## 5. EA as Template Antioxidant Molecule for the Development of New Therapeutics for AD

EA attracts the interest of researchers as a promising molecule to provide benefits in neurodegenerative disorders, including AD, mainly due to its anti-inflammatory and antioxidant properties. Defining which pharmacophore/pharmacophores in EA are actually responsible/s for its health benefits and its possible collateral effects is crucial for in silico screening investigations and designing new multi-target EA-type CNS drugs. The mechanisms at the basis of the EA multifaceted bioactivity are mainly based on its antioxidant, radical scavenger, and anti-aging effects, capable of contrasting OS. Collectively, EA is capable of counteracting the detrimental RONS, which are a byproduct of the physiologic aerobic metabolism. For a more precise distinction, OS refers to a torrent of destructive proceedings that frequently trigger and accompany the molecular/cellular pathogenic events responsible for several human disorders, including AD [144,195]. Differently, inflammation, being both the cause and the effect of RONS accumulation, is considered a pathological characteristic of most human diseases, including those developing in the CNS, including AD.

### 5.1. Antioxidant Effects of EA: Proposed Mechanisms of Action

Natural antioxidants are fundamentally present in vegetable food, and polyphenols, such as EA, are supposed to comprise more than 8000 molecules, all characterized by possessing at least a phenol moiety. EA hydroxyl groups and the lactone systems give the molecules the capacity to form hydrogen bonds and can also act as electron acceptors and/or hydrogen donors. Consequently, EA is endowed with the capacity to take electrons from different substrates, thus promoting antioxidant redox reactions and functioning as a very efficient free radical (FR) scavenger [196]. The EA anion is proposed as the key species for its protective effects against OS [196]. It is predicted to be efficiently and continuously regenerated after scavenging two free radicals per cycle [196]. Chemical species able to prevent oxidation can be classified into primary antioxidants (Type I, or chain-breaking) and secondary antioxidants (Type II, or preventive). EA can behave as both Type I and Type II antioxidants, thus exerting multiple-function antioxidant activity (Table 9) [197].

#### 5.1.1. Type I Scavenging Reactions

Type I scavenging reactions, which can occur between EA and FRs, follow second-order kinetics and scavenging capacity, as well as its velocity, depending both on the concentration of EA and FRs. Factors that could modify their chemical structures, such as the pH, polarity, reaction conditions, and medium, could also affect EA scavenging capacity. In general, the antioxidant capacity of EA reduces strongly in solvents able to form hydrogen bonds with EA and improve in solvents, favouring EA ionization to anion phenoxide [198]. The alcohols may act as acceptors of hydrogen bonds, thus decreasing EA antiradical effects via hydrogen atom transfer (HAT) reactions. On the other hand, they can favour the ionization of the EA to anion phenoxides, which can react rapidly with peroxyl radicals through electron transfer, thus improving EA radical scavenging activity via SET reactions [198]. In general, the antiradical properties of different natural and synthetic Type I antioxidants possessing OH groups mainly derive from their capacity to transfer hydrogen atoms to FRs. This process can occur via the different mechanisms reported in Table A2 (Appendix A). These mechanisms generate non-radical species or new radicals more stable and less reactive than the previous ones, thus restricting the development of OS. Table A2 also reports the chemical equations associated with these proposed mechanisms. EA can mainly exercise antioxidant effects through three of the reaction mechanisms reported in Table A2, such as SET, HAT, and SPLHAT reactions. Although the result is always the inactivation of FRs to neutral, cationic, or anionic species, the kinetics and secondary reactions involved in the processes are different (Figure A4, Appendix A).

When EA reacts, for example, with the radical species ROO•, a hydrogen cation coming from its hydroxyls into other radical species is transferred, forming a transition state of an H-O bond with one electron. On the other hand, the hydroxyl groups can interact with the π-electrons of the benzene ring, providing molecules endowed with the ability to generate free long-living radicals stabilized by delocalization and able to interfere and modify radical-mediated oxidation processes via SET reactions.

#### 5.1.2. Type II Scavenging Reactions

EA is also a Type II antioxidant, thus providing protective effects against FRs by inhibiting the endogenous production of oxidants and radical hydroxyl (•OH) molecule, which is the most reactive and electrophilic species of oxygen-based radicals [30]. •OH is mainly responsible for tissues and DNA damage and, therefore, its inhibition is of prime significance for reducing OS generated from the metal-catalysed Fenton reaction and Haber Weiss recombination (HWR), according to Equations (1)–(4), involving the reduced forms of Fe and Cu.Fe (II) + H_2_O_2_ → Fe (III) + OH^−^ + •OH(1)Cu (I) + H_2_O_2_ → Cu (II) + H^−^ + •OH(2)Fe (III) + O_2_^•−^ → Fe (II) + O_2_(3)Cu (II) + O_2_^•−^ → Cu (I) + O_2_ (Fenton)(4)

In this context, EA is an excellent antioxidant due to its capability to chelate and subtract metal such as Fe^2+^, Fe^3+^, and copper ions involved in the production of FRs, thus preventing the oxidation of low-density lipoproteins (LDL) [196,197,199]. EA can also interact with enzymes involved in radical generation, such as various cytochrome P450 isoforms, lipoxygenases, cyclooxygenase, and xanthine oxidase, thus inhibiting RONS over production. This capability derives from the presence of the hydrophobic benzenoid rings and from the skill of the phenolic hydroxyl groups to form hydrogen-bonding interactions [200]. Moreover, EA can act synergistically with other endogenous and exogenous antioxidants, such as ascorbic acid, β-carotene, and β-tocopherol, thus increasing their effectiveness and regulating intracellular glutathione levels [200]. Unfortunately, some of the hydroxyl groups of EA, in conditions of high dosage, high concentrations of transition metal ions, alkali pH, and/or the presence of oxygen molecules, can also act unexpectedly as pro-oxidant moieties [201]. These groups may sometimes induce significant DNA damage in the presence of Cu (II) or may create ROS through the reduction of Cu (II)→Cu. The pro-oxidant activity is peculiar of small polyphenols, such as EA, but is limited in large-molecular-weight phenols, such as ETs. On the other hand, this apparent issue can trigger apoptosis in cancer cells [202,203].

## 6. EA-Rich Foods, EA Food Supplements, and EA Involvement in the Treatment of AD

As above-mentioned, the polyphenolic lactone with the formula C_14_H_6_O_8_, known as EA, as well as the intake of EA food supplements and foods rich in ETs and/or EA can translate into altering profuse signaling inside cells, thus preventing and/or pauperizing the progression of diverse neurodegenerative abnormalities, including AD [204]. Its neuroprotective effectiveness is mainly attributable to its ROS scavenging, iron chelating properties, positive regulation of energetics of mitochondrial respiratory complex, and abundant modulation of neuronal molecular signaling pathways [205].

### Most Relevant In Vitro and In Vivo Studies Using ETs and EA-Rich Plants

Table 10 summarizes the beneficial properties demonstrated in vitro and/or in vivo studies using different experimental models, or even in clinical settings, observed upon the assumption of ETs and EA-rich plants.

Given the information reported in Table 10, it appears unequivocally that the clinical interest in the possible beneficial properties of EA-rich plants is very limited. Particularly, among the studies considered here (56), the clinical ones represent only 5%, and in vivo ones largely comprise under half a percent (25%) of the in vitro ones (70%) (Figure A5, Appendix A). Collectively, practically all studies, regardless of whether they were conducted in vitro, in vivo, or in clinical settings, mainly revealed antioxidants and anti-inflammatory effects.

Although among the considered studies, a neuroprotective action was mentioned in only one case [238], as already extensively claimed in this review, inflammation and OS evidenced in all other studies are detrimental processes pivotal in the onset and development of AD, thus confirming the high potentialities of EA and EA-rich plants to at least prevent AD arrival. However, other in vitro studies exist reporting on the neuroprotective effects of *Punica granatum* [257] and *Cochlospermum. angolensis* bark extracts [242]. The administration of *P. granatum* reduced Aβ deposition via a specific non-competitive inhibition of BACE1 activity [257]. Bark extracts exerted potent radical scavenging activity, thus limiting OS and reducing cholinesterase activities while potentiating monoaminergic functions by reducing MAO activity and preserving biogenic amines [242]. Moreover, the in vivo administration of 6.25 mL/L of pomegranate extracts (POMx) in the drinking water for 3 months [258] to C57BL/6 APPswe/PS1dE9 transgenic mice (male) reduced microgliosis and AD progression and improved spatial learning, motor functions, memory performance, and behavioural performance by decreasing the concentration of TNF-α, NFAT, and cytokines; reducing Aβ production and IkB degradation; and inhibiting the production of NF-kB. Similarly, the administration of 6.25 mL/L of pomegranate juice (PJ) in the drinking water for 6 to 12.5 months of age to C57BL/6 APPsw/Tg2576 trans-genic mice (male) reduced amyloid deposition in the hippocampus and improved learning and memory abilities, motor functions, and behavioural performance with dipping Aβ42 concentrations [259]. Table 11 reports the results of quantitative analyses of the ETs and EA content in various fruits, nuts, and beverages. It is important to know that among ET-rich food as an in vivo source of EA, punicalagin (found predominantly in pomegranate) sanguiin H-6 in strawberry and raspberry, vescalagin in oak-aged wines and spirits, and pedunculagin in walnuts are the ETs providing the highest amounts of EA.

Despite its very low bioavailability, more interest was demonstrated in the evaluation of the effects of isolated EA both on stressors associated with AD and on AD symptoms. Table 12 reports some relevant in vitro studies that revealed the effects of isolated EA against several stressors found in AD and/or recognized as engaged in the onset and development of AD.

In addition, the administration in vitro of commercial EA was able to decrease the oxidative DNA damage and free radical concentration [268,275] by limiting dopamine oxidation and the concentrations of neurotoxins, oxygen superoxide, and H_2_O_2_ and exerting potent radical scavenging activity. Additionally, a reduction in AChE activity detrimental to AD was observed [268]. Another study reported that EA administration reduced the production and toxicity of Aβ oligomers by decreasing Aβ oligomerization, soluble Aβ42 levels, and Aβ42 toxicity in SH-SY5Y neuroblastoma cells used as in vitro model [267]. Also, EA in vitro administration was able to improve monoaminergic functions by reducing MAO-A activity [242].

Table 13 and Table 14 summarize the in vivo assessment of the neuroprotective effects of EA in various AD animal models and animal models of pathologic conditions present in AD development. Specifically, in Table 13, the biomarkers evaluated, and the positive variations observed in the pathology processes are included, and the involved mechanisms of action of EA are included in Table 14.

It is universally recognized that inflammation and OS are pivotal to the onset and development of the clinical signs and the pathological hallmarks that typify AD [14]. Increased levels of proinflammatory cytokines such as TNF-α, IL-1β, IL-6, and interferon g (IFN-g) reduce the Aβ phagocytosis in the AD-affected brain, interfering with the physiological mechanisms of plaque removal and then worsening astrogliosis and neural death, supporting the progression of the disease [14,17]. On the other hand, the overaccumulation of RONS and the development of OS, caused by metal ion imbalance, contribute to the development and progression of AD. Specifically, they promote amyloid-β (Aβ) overproduction, cause τ hyperphosphorylation, disrupt organelles, and cause endoplasmic reticulum (ER) stress and mitochondrial and autophagic dysfunctions, which impair synaptic functions, thus leading to chronic neurodegeneration and cognitive deficits, such as those seen in AD patients [303]. Other abnormalities observable in CNS, including malondialdehyde and 4-hydroxy-2-nonenal altered levels, increased lipid peroxidation, and pervasive protein oxidation, determine high levels of nitro-tyrosine and increased amounts of 8-hydroxy-2-deoxyguanosine link OS to AD [304]. Even if adjustment of metal balance by supplementing chelators of the metal ions may have potential in ameliorating AD pathologies, the possible therapeutic benefits of dietary multifaceted molecules such as EA capable of both contrast inflammation and OS in AD have been and are currently under intense investigation. It has been reported that in vitro, EA from *Punica granatum* inhibits the activity of the b-secretase (BACE1), a cleaving enzyme involved in the production of Aβ from amyloid precursor protein (APP), with relative specificity [257]. Accordingly, in vivo administration of pomegranate juice (which is particularly enriched in EA and punicalagin, a source of EA) to APP/PS1 transgenic mice, an animal model of AD, elicited a significant amelioration in spatial learning and motor functions and a marked reduction in the endogenous level of Aβ peptide (Aβ42), TNF-a, NFAT, and microgliosis in the hippocampus [258,259]. Although apparently in contrast with such results, Feng and colleagues also concluded that EA could be neuroprotective in patients suffering from AD because of its ability to promote endogenous mechanisms of protection aimed at reducing the bioavailability of the soluble form of Aβ protein in the bio-phase [267]. Kiasalari and co-workers confirmed that in vivo, EA ameliorates behavioural skills and neuronal defects, provoked by the microinjection of Aβ peptide in the CNS [300]. The anti-inflammatory and antioxidant properties of EA were further confirmed in a Streptozotocin (STZ) intra-cerebral injected animal model of AD (SAD rats), which developed detrimental hallmarks that mimic those observed in the sporadic form of AD [268]. The in vivo EA treatment in these animals revealed a marked reduction in AChE activity paralleled by the restoration of the synaptic pool of ACh. EA also caused a significant reduction in Aβ deposition, reduced OS, and neural apoptosis. Summing up, although further studies are needed to confirm the hypothesis of the neuroprotective action of EA in AD, the results from both in vitro and in vivo experiments assert rational justifications for looking to EA as a compound of great interest for potential applications as a memory restorative agent in the treatment of dementia and AD [268]. Finally, in a relatively recent study by our colleagues, it has been demonstrated that the oral administration of a new oral EA micro-dispersion (EAm), with increased EA solubility, although it did not modify animal weight and behavioral skills, significantly recovered changes in “ex-vivo, in vitro” parameters in old animals when compared to young ones [190]. Moreover, EAm treatment significantly reduced the CD45 signal in both young and old cortical lysates, and it diminished GFAP immunopositivity in young mice. Finally, EAm treatment significantly reduced IL1β expression in old mice. These results suggest that EAm benefits aging and represents a nutraceutical ingredient for elders [190].

## 7. Conclusions, Perspectives for the Future, and Authors’ Opinions

Currently, available dementia services worldwide are inadequately resourced and staffed, mainly community-based, and highly fragmented. On the contrary, multidisciplinary teams and facilities will be needed to correctly and safely administer all new therapies that are arising for AD, and their correct delivery will require an accurate molecular diagnosis of AD. In the UK, only about 60% of people potentially with dementia receive even a clinical diagnosis of dementia. Despite the guidance from the National Institute for Health and Care Excellence recommends structural imaging, there is wide variation in imaging use between centres.

### 7.1. Imaging Analyses Available to Confirm the Presence of AD

There is wide variation in the proportion of patients receiving a scan. More worryingly, among people who have a scan, the majority had only a computed tomography (CT) scanning of the head, which combines special X-ray equipment with sophisticated computers to produce multiple images or pictures of the brain to look for and rule out other causes of dementia, such as a brain tumor, subdural hematoma, or stroke, with only 26% having an MRI. Specifically, magnetic resonance imaging (MRI) uses a powerful magnetic field, radio frequency pulses, and a computer to produce detailed pictures that can detect brain abnormalities associated with mild cognitive impairment (MCI) and can be used to predict which patients with MCI may eventually develop AD. Although in the early stages of AD, an MRI scan of the brain may be normal, in later stages, an MRI may show a decrease in the size of different areas of the brain (mainly affecting the temporal and parietal lobes). Moreover, less than 2% of patients receive molecular confirmation of their disease using CSF biomarkers, as included in NICE guidance, or an amyloid positron emission tomography (PET) scan analysis, which is a diagnostic examination that uses small amounts of radioactive material (called a radiotracer) to diagnose and determine the severity of a variety of diseases. A combined PET/CT exam fuses images from a PET and CT scan together to provide detail on both the anatomy (from the CT scan) and function (from the PET scan) of the brain. A PET/CT scan can help differentiate Alzheimer’s disease from other types of dementia. Another nuclear medicine test called a single-photon emission computed tomography (SPECT) scan could also be used for this purpose. Additionally, using PET scanning and a new radiotracer called C-11 PIB, scientists have recently imaged the build-up of beta-amyloid plaques in the living brain. Radiotracers similar to C-11 PIB are currently being developed for use in the clinical setting.

### 7.2. An Opportunity to Change

Although NICE guidelines are not available for the investigation and management of people with mild cognitive impairment, the advent of new therapies provides an opportunity for change. The recent availability of disease-modifying drugs for AD might bring an influx of people into clinical services, including those with AD, those with other dementias, and individuals concerned about their risk of developing dementia and/or AD. Clear referral criteria and equitable pathways from primary care to specialist services will be required. Access must not be limited to those living near specialist centres, and health systems must also ensure access for minorities and individuals living alone. “Time is brain” should be adopted. Diagnostic delays for AD might adversely affect outcomes of the new disease-modifying therapies. If disease progression can be slowed, then initiating treatment as early as possible could result in maximal benefit. The clinical implementation of these new drugs will, at least initially, likely resemble the methodology used in clinical trials. Greater access to diagnostic tests will be required, and demand for MRI could be a major bottleneck. It is likely that more scanners will be needed, and a more efficient use of existing scanners, including the development of shorter, focused protocols and neuroradiological expertise for scan interpretation and the detection of amyloid-related imaging abnormalities (ARIA).

## Data Availability

Not necessary.

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
