# Peer review of "Ellagic Acid: A Green Multi-Target Weapon That Reduces Oxidative Stress and Inflammation to Prevent and Improve the Condition of Alzheimer’s Disease"

_ijms, 2025, doi:10.3390/ijms26020844_

Round 1

Reviewer 1 Report

Comments and Suggestions for Authors

In the manuscript (ijms-3400708), the authors reviewed the studies on the prevention and improvement of Alzheimer's disease (AD) by ellagic acid (EA) in terms of reducing oxidative stress and inflammation. In general, the contents meet the requirements of International Journal of mechanical Sciences. However, there remain some issues that still need to be corrected before the manuscript can be accepted for publication in International Journal of mechanical Sciences. To ensure the publication readiness of the manuscript, the following issue needs to be addressed:

(1) Title: Please change “A Green Multi-Target Weapon Reducing Oxidative Stress and Inflammation Thus Preventing and Ameliorating Alzheimer Disease (AD) Condition” to “A Green multi-target weapon that Reduces Oxidative Stress and Inflammation to Prevent and Improve the Condition of Alzheimer's Disease (AD)”.

(2) Line 17: Please change “antioxidants” to “antioxidant”.

(3) 1. Introduction: This review lacks the description of the writing method, such as which keywords the author chose to consult the literatures, which databases were used, the number of literatures collected and the number of literatures finally selected, etc. Suggest authors to add the method.

(4) Line 47: Please change “dementia[2]” to “dementia [2]”. In addition, there are similar problems in other parts of the manuscript, such as line 62, 64, and 106. Therefore, authors are advised to check the whole manuscript carefully and correct these minor problems.

(5) In general, the quality and aesthetics of all the figures in this review are not high, and it is suggested that the author improve the quality of the diagrams.

(6) Line 123-124: ellagitannins (ETs) as well as gallic acid (GA) and ellagic acid (EA). It is recommended that authors draw their chemical structures in Figure 2, so that readers can read and understand the content more easily.

(7) Line 141: Please change “radical scavenging and antioxidant capacity of EA were” to “radical scavenging and antioxidant capacity of EA was”.

(8) Line 159: "in vivo" should be italicized.

(9) Line 167-169: Such NDs comprehend, among others, Alzheimer’s disease (AD), Parkinson’s disease (PD), and Huntington’s disease (HD), as well as amyotrophic lateral sclerosis (ALS). This sentence does not follow logic, suggest that the authors reorganize it.

(10) Table 1: The lack of borders in Table 1 causes the content to be confusing. Please modify it carefully.

(11) Figure 5 is too vague, the authors are advised to modify it carefully.

(12) Line 356: Please change “Current AD Therapies” to “3.1.1. Current AD Therapies”.

(13) Table 5: Table 5 is too large for the reader to read; It is suggested that the author divide it into smaller tables according to the content. In addition, there are similar problems in other table of the manuscript, and authors are advised to check the whole manuscript carefully and correct these problems.

(14) Line 418: Please change “Multi-Target Therapy (MTT) for AD” to “3.1.2. Multi-Target Therapy (MTT) for AD”.

(15) Line 646: Please change “5.1. EA Antioxidant Effects: Proposed Mechanisms of Action” to “5.1. Antioxidant Effects of EA: Proposed Mechanisms of Action”.

(16) Line 664: Please change “Type I scavenging reactions” to “5.1.1. Type I Scavenging Reactions”. It is recommended that authors carefully read the submission requirements of International Journal of mechanical Sciences, and confirm whether the first letter of each word in the title needs to be capitalized, and unify the writing of the words in the title.

(17) Line 695: Please change “Type II scavenging reactions” to “5.1.2. Type II Scavenging Reactions”.

Author Response

In the manuscript (ijms-3400708), the authors reviewed the studies on the prevention and improvement of Alzheimer's disease (AD) by ellagic acid (EA) in terms of reducing oxidative stress and inflammation. In general, the contents meet the requirements of International Journal of mechanical Sciences. However, there remain some issues that still need to be corrected before the manuscript can be accepted for publication in International Journal of mechanical Sciences. To ensure the publication readiness of the manuscript, the following issue needs to be addressed:

We thank a lot the Reviewer for his/her careful and useful work of revision.

(1) Title: Please change “A Green Multi-Target Weapon Reducing Oxidative Stress and Inflammation Thus Preventing and Ameliorating Alzheimer Disease (AD) Condition” to “A Green multi-target weapon that Reduces Oxidative Stress and Inflammation to Prevent and Improve the Condition of Alzheimer's Disease (AD)”

Dear Reviewer, thank you for your suggestion. I agree with it. Therefore, the title has been changed as asked. Please, see lines 2-4.

(2) Line 17: Please change “antioxidants” to “antioxidant”.

The suggested modification has been applied. Please, see line 20.

(3) 1. Introduction: This review lacks the description of the writing method, such as which keywords the author chose to consult the literatures, which databases were used, the number of literatures collected and the number of literatures finally selected, etc. Suggest authors to add the method.

As suggested by the reviewer whom we thank, the methods used to edit this review have been added as Section 1.1 at the beginning of the introduction. Please see lines 47-66.

(4) Line 47: Please change “dementia[2]” to “dementia [2]”. In addition, there are similar problems in other parts of the manuscript, such as line 62, 64, and 106. Therefore, authors are advised to check the whole manuscript carefully and correct these minor problems.

We thank a lot the Reviewer for his/her suggestion. By checking all the manuscript, we have found several issues like those indicated by the Reviewer, which have been corrected. Anyway, the problem is mainly due to the use of Mendeley software to insert references.

(5) In general, the quality and aesthetics of all the figures in this review are not high, and it is suggested that the author improve the quality of the diagrams.

Although all the Figures already respected the quality requirements of IJMS, we further improved their resolutions. Additionally, the type of diagrams is that offered by the power point software, and concerning our previous works on different MDPI journal, they were appreciated and accepted.

(6) Line 123-124: ellagitannins (ETs) as well as gallic acid (GA) and ellagic acid (EA). It is recommended that authors draw their chemical structures in Figure 2, so that readers can read and understand the content more easily.

We thank a lot the Reviewer for his/her suggestion with which we agreed. Therefore, Figure 2 has been modified by inserting the chemical structures of ellagitannins and GA.

(7) Line 141: Please change “radical scavenging and antioxidant capacity of EA were” to “radical scavenging and antioxidant capacity of EA was”.

The asked modification has been applied. Please, see at lines 171-172.

(8) Line 159: "in vivo" should be italicized.

The “in vivo” voice signalled by the Reviewer has been italicized as asked. Please, see line 220.

(9) Line 167-169: Such NDs comprehend, among others, Alzheimer’s disease (AD), Parkinson’s disease (PD), and Huntington’s disease (HD), as well as amyotrophic lateral sclerosis (ALS). This sentence does not follow logic, suggest that the authors reorganize it.

As asked by the Reviewer, the sentence has been reorganized according to Neurodegenerative disease - Wikipedia. Please, see lines 228-231.

(10) Table 1: The lack of borders in Table 1 causes the content to be confusing. Please modify it carefully.

As asked by the Reviewer, additional vertical borders have been inserted in Table 1.

(11) Figure 5 is too vague, the authors are advised to modify it carefully.

We thank the Reviewer a lot for the comment, but since the same Figure originally prepared by us in 2020 has been considered clear and accepted by two MDPI relevant journals such as Antioxidants and the same IJMS, we are confident that it is sufficiently explanatory and clear, as such. Indeed, it contains enough details to enable readers to understand the pathways of ROS production and the main effects which they have on biological systems.

(12) Line 356: Please change “Current AD Therapies” to “3.1.1. Current AD Therapies”.

The numbering of the indicated Section has been inserted (line 415). The same has been applied to the following two Sections 3.1.2 (line 459) and 3.1.3. (line 473).

(13) Table 5: Table 5 is too large for the reader to read; It is suggested that the author divide it into smaller tables according to the content. In addition, there are similar problems in other table of the manuscript, and authors are advised to check the whole manuscript carefully and correct these problems.

The Reviewer is right. Anyway, for experience, the formatting issues, appearance and style of Table are usually solved by the Editorial Office after acceptance of articles and before proofreading. In this regard, it is not convenient to modify them at this early phase.

(14) Line 418: Please change “Multi-Target Therapy (MTT) for AD” to “3.1.2. Multi-Target Therapy (MTT) for AD”.

As reported in my answer to point (12), the operation requested has been already done, but the correct numbering is 3.1.3., due to the presence of a previous sub-section that has been numbered 3.1.2.

(15) Line 646: Please change “5.1. EA Antioxidant Effects: Proposed Mechanisms of Action” to “5.1. Antioxidant Effects of EA: Proposed Mechanisms of Action”.

The change requested by the Reviewer has been applied (line 704). Additionally, the numbering of the following two sections (5.1.1., line 724, and 5.1.2., line 752) has been inserted to follow previous suggestions by the Reviewer.

(16) Line 664: Please change “Type I scavenging reactions” to “5.1.1. Type I Scavenging Reactions”. It is recommended that authors carefully read the submission requirements of International Journal of mechanical Sciences, and confirm whether the first letter of each word in the title needs to be capitalized, and unify the writing of the words in the title.

(17) Line 695: Please change “Type II scavenging reactions” to “5.1.2. Type II Scavenging Reactions”.

As the Reviewer can observe in my answer to point (15), the numbering of the sub-sections signaled in points (16) and (17) were already inserted. Additionally, we checked carefully all the manuscript to assure that the first letter of each word in the titles is capitalized, and the writing of the words in the title has been unified. Sorry but the correct name of the journal IJMS is International Journal of Molecular Sciences.

Reviewer 2 Report

Comments and Suggestions for Authors

The text provides a comprehensive perspective on oxidative stress (OS) and its relationship with Alzheimer’s disease (AD), focusing on the potential therapeutic role of polyphenols such as ellagitannins (ETs), ellagic acid (EA), and urolithins (UROs).

  • The review highlights the potential of polyphenols like ETs, EA, and UROs as eco-friendly alternatives or adjuvants to traditional drugs.
  • The analysis does not shy away from addressing challenges related to the bioavailability of ETs and EA, as well as the variability in UROs formation, offering a balanced perspective.
  • The text integrates discussions on pathophysiology, chemical properties, antioxidant mechanisms, and potential clinical applications, making it well-structured and informative.
  • It includes detailed descriptions of how EA and UROs counteract oxidative stress, such as interactions with specific enzymes (e.g., superoxide dismutase, catalase) or signaling pathways (e.g., inhibition of NF-κB, activation of Nrf2).
  • The authors have added diagrams and tables to summarize the mechanisms of action, challenges, and therapeutic potential of ETs, EA, and UROs for improved clarity and engagement.
  • The bibliography is appropriately selected, and the information is well-structured.

Comments:

A1. I sugest using more concise keywords.

A2. Pay attention to punctuation. For example, add a period at the end of line 629.

A3. Figure 7 should be revised; the color contrast used makes it difficult to understand.

A4. I suggest adding a subchapter where the authors detail how the bibliography was selected and how the study was conceptualized.

Author Response

The text provides a comprehensive perspective on oxidative stress (OS) and its relationship with Alzheimer’s disease (AD), focusing on the potential therapeutic role of polyphenols such as ellagitannins (ETs), ellagic acid (EA), and urolithins (UROs).

  • The review highlights the potential of polyphenols like ETs, EA, and UROs as eco-friendly alternatives or adjuvants to traditional drugs.
  • The analysis does not shy away from addressing challenges related to the bioavailability of ETs and EA, as well as the variability in UROs formation, offering a balanced perspective.
  • The text integrates discussions on pathophysiology, chemical properties, antioxidant mechanisms, and potential clinical applications, making it well-structured and informative.
  • It includes detailed descriptions of how EA and UROs counteract oxidative stress, such as interactions with specific enzymes (e.g., superoxide dismutase, catalase) or signaling pathways (e.g., inhibition of NF-κB, activation of Nrf2).
  • The authors have added diagrams and tables to summarize the mechanisms of action, challenges, and therapeutic potential of ETs, EA, and UROs for improved clarity and engagement.
  • The bibliography is appropriately selected, and the information is well-structured.

Comments:

A1. I sugest using more concise keywords.

As suggested, original keywords have been modified. More concise keywords have been used. Please, see lines 41-44.

A2. Pay attention to punctuation. For example, add a period at the end of line 629.

As suggested, a period have been inserted where required by the Reviewer. Please, see line 687 (revised manuscript).

A3. Figure 7 should be revised; the color contrast used makes it difficult to understand.

We thank a lot the Reviewer for his/her useful suggestion. Accordingly, both content, the graphic and colours of original Figure 7 have been completely modified. Now, on suggestion of another Reviewer, Figure 7 has been moved to the new Appendix B, where it appears as Figure 1B.

A4. I suggest adding a subchapter where the authors detail how the bibliography was selected and how the study was conceptualized.

On suggestion of Reviewer 1, the required subchapter has been already inserted at the beginning of the introduction. Please, see lines 47-66.

Reviewer 3 Report

Comments and Suggestions for Authors

1.  Abstract Concerns: The abstract mentions three concepts: ellagitannins (ETs), ellagic acid (EA), and urolithins (UROs). The paper's main focus is EA, but based on the abstract, “Unfortunately, in vivo ETs are rather not absorbed, while providing mainly ellagic acid (EA), which due to its trivial water-solubility, first pass effect… thus failing as therapeutic in vivo,” it appears that ETs are ineffective. Furthermore, “Up-to-date, only UROs have confirmed the beneficial effect demonstrated in vitro” suggests that UROs have proven efficacy. This raises a key question: why is the study focusing on EA instead of UROs? The rationale for choosing EAs as the central theme is unclear.

2.  Introduction – Abrupt Transition: In the Introduction, the statement “It is anyway extensively recognized that OS, as well as the formation of free radical and non-radical RONS, are strongly involved in the progression of brain aging and, in the onset, and evolution of AD” appears abrupt. The preceding discussion focuses on other mechanisms of Alzheimer’s disease (AD) pathogenesis, such as excessive acetylcholine esterase enzymes (AChE) and β-amyloid precursor protein-cleaving enzyme 1 (BACE-1). Introducing oxidative stress (OS) here feels disjointed and lacks a logical flow.

3.  Introduction – Logical Inconsistency: In the third paragraph of the Introduction, it is stated that “As abovementioned, ETs are capable to provide EA by hydrolysis, which is rationally considered the bioactive fragment of ETs possessing one of the strongest antioxidant powers capable to counteract OS.” However, based on the earlier discussion, one cannot deduce that ETs provide EA; instead, the claim only underscores the importance of ETs in AD treatment. The logic here is inconsistent.

4.  Introduction – Lack of Evidence for Studying EA: The third paragraph of the Introduction does not provide a clear rationale for studying EA. Instead, it emphasizes the significance of ETs, raising the question of why ETs are not the primary focus of the research. Given that ETs produce various metabolites, the justification for singling out EA is insufficient. Furthermore, there is no direct evidence supporting EA’s efficacy in treating AD, such as results from animal, cell, or clinical studies. The lack of evidence undermines the case for prioritizing EA in this study.

5.  Tables – Irrelevance to the Theme: The paper includes 15 tables, but only Table 15 directly relates to the theme of EA’s potential for treating AD. The other tables focus on either other drugs for AD treatment or EA’s effects on unrelated diseases, such as cancer. This diminishes the paper’s focus on its stated theme.

6.  Structure and Theme Misalignment: Of the eight sections in the paper, only Section 6 aligns with the stated theme, and even then, it addresses AD treatment mechanisms rather than the theme of "Reducing Oxidative Stress and Inflammation." The lack of focus is evident, as only 1/8 of the content is tangentially related to the theme.

Author Response

Abstract Concerns: The abstract mentions three concepts: ellagitannins (ETs), ellagic acid (EA), and urolithins (UROs). The paper's main focus is EA, but based on the abstract, “Unfortunately, in vivo ETs are rather not absorbed, while providing mainly ellagic acid (EA), which due to its trivial water-solubility, first pass effect… thus failing as therapeutic in vivo,” it appears that ETs are ineffective. Furthermore, “Up-to-date, only UROs have confirmed the beneficial effect demonstrated in vitro” suggests that UROs have proven efficacy. This raises a key question: why is the study focusing on EA instead of UROs? The rationale for choosing EAs as the central theme is unclear.

We can understand the concerns raised by the Reviewer, but in a subsequent sentence in the abstract, it has been reported the reason for which UROs, despite they better bioavailability, are not considered as possible therapeutic option to treat AD or other diseases, thus confirming EA as the lead-compound to be promising for this purpose, upon due formulation studies. Please, consider the following part of the abstract.

Unfortunately, upon administration of food rich in ETs or ETs and EA, UROs formation is affected by extreme interindividual variability that renders them unreliable as novel clinically usable drugs. Large attention has been therefore paid specifically to multitarget EA, which is incessantly investigated as such or nanotechnologically manipulated to be a potential “lead compound” with protective action towards AD”.

Additionally, Section 4 (already present in the original not revised manuscript) should be sufficient at further addressing the Reviewer “key question”. Specifically, the answer at the key question raised by the Reviewer “why is the study focusing on EA instead of UROs?” is present and clearly expressed both in the abstract and in Section 4.3. Similarly, the rational for choosing EA as the central theme of this Review is further extensively clarified in the same Section 4.3. It is also the reason for which several other researchers before us have focused their attention and experimental studies on EA rather than on UROs (or ETs). Please, reconsider better all the abstract and Section 4.

  1.  Introduction – Abrupt Transition: In the Introduction, the statement “It is anyway extensively recognized that OS, as well as the formation of free radical and non-radical RONS, are strongly involved in the progression of brain aging and, in the onset, and evolution of AD” appears abrupt. The preceding discussion focuses on other mechanisms of Alzheimer’s disease (AD) pathogenesis, such as excessive acetylcholine esterase enzymes (AChE) and β-amyloid precursor protein-cleaving enzyme 1 (BACE-1). Introducing oxidative stress (OS) here feels disjointed and lacks a logical flow.

Why does the Reviewer consider talking about RONS and OS as an abrupt transition. Like excessive acetylcholine esterase enzymes (AChE) and β-amyloid precursor protein-cleaving enzyme 1 (BACE-1) and many other factors reported in the text, also RONS and OS are strongly involved in the progression of brain aging and, in the onset, and evolution of AD. In this part of the Introduction, we aimed at providing a series of the most recognized causes of the onset and development of AD, which is in fact a multifactorial and very complex disease, whose exact origin and progression are not still well understood. Together with genetic reasons, AD is a well-unshakable neuronal disfunction, whose primary causes could be related to several motives. Toxin insults, heredity, metabolism, attack by infectious pathogens, amyloid plaques, neurofibrillary tangles found in the brain, excessive acetylcholine esterase enzymes (AChE), β amyloid (βA) precursor protein-cleaving enzyme 1 (BACE-1), glycogen synthase kinase 3 β (GSK-3 β), monoamine oxidases (MAOs), metal ions in the brain, N-methyl-D-aspartate (NMDA) receptors, phosphodiesterase (PDE), impaired bioenergetics, mitochondrial abnormalities, neuroinflammatory, as well as RONS and OS, are among the most recognised causes of AD. In this regard, like other above-reported factors, RONS and OS participate to the onset and development of AD, as well as of the most part of human diseases. Additionally, it seems that RONS and OS are responsible of the increased production of toxic Aβ, mentioned above and with which they are logically connected. We are sure that, rather than an abrupt transition, introducing OS and RONS among the other causes of AD pathogenesis gives a rational and logical flow to the disquisition. Figure 1, which reports among other endogenous factors and possible biological targets involved in AD pathology, oxidative stress (caused by RONS hyperproduction), is a confirmation that talk about RONS and OS in sequence to the other AD caused, is logical and rational.

  1. Introduction – Logical Inconsistency: In the third paragraph of the Introduction, it is stated that “As abovementioned, ETs are capable to provide EA by hydrolysis, which is rationally considered the bioactive fragment of ETs possessing one of the strongest antioxidant powers capable to counteract OS.” However, based on the earlier discussion, one cannot deduce that ETs provide EA; instead, the claim only underscores the importance of ETs in AD treatment. The logic here is inconsistent.

I apologise in advance with the Reviewer, but perhaps he/she has misinterpreted the sentence that she/he has reported in point 3 or even all the dissertation on ETs and EA. If I have understood correctly the Reviewer comment, he/she asserts that the sentence reported in point 3 would underscore the importance of ETs in AD treatment rather than that of EA. Actually, it means the opposite. As more in deep explained also in the new part added in lines 182-200 and in the core of manuscript, although also ETs possess radical scavenging and antioxidant effects, these are due mainly to the presence in their structure of units of EA, and are however inferior to those of EA. Nonetheless, ETs are not absorbed but, upon hydrolysis in the intestine provide free EA, which can develop its antioxidant powers capable to counteract OS. The superior antioxidants effect of free EA could better help in the treatment of AD. I hope to have better elucidated the concepts. Anyway, for more clarity, the sentence has been slightly modified. Please, see lines 168-171.

  1. Introduction – Lack of Evidence for Studying EA: The third paragraph of the Introduction does not provide a clear rationale for studying EA. Instead, it emphasizes the significance of ETs, raising the question of why ETs are not the primary focus of the research. Given that ETs produce various metabolites, the justification for singling out EA is insufficient. Furthermore, there is no direct evidence supporting EA’s efficacy in treating AD, such as results from animal, cell, or clinical studies. The lack of evidence undermines the case for prioritizing EA in this study.

A clear rationale for studying EA, rather than ETs or other ETs metabolites, as well as EA metabolites (UROs), thus leaving EA as the primary focus of the research, has been provided (lines 182-200). Concerning the existence of direct evidence supporting EA’s efficacy in treating AD, such as results from animal, cell, or clinical studies they are described in Section 6.1 and specifically in vitro and in vivo results were collected in the original Table 14, 15 and 16, now Table 12, 13 and 14. So the Reviewer comment “The lack of evidence undermines the case for prioritizing EA in this study” loses credibility.

  1. Tables – Irrelevance to the Theme: The paper includes 15 tables, but only Table 15 directly relates to the theme of EA’s potential for treating AD. The other tables focus on either other drugs for AD treatment or EA’s effects on unrelated diseases, such as cancer. This diminishes the paper’s focus on its stated theme.

It is curious that the Reviewer has noticed Table 15 (Table 13 in the revised manuscript) and in the previous point has asserted that “there is no direct evidence supporting EA’s efficacy in treating AD, such as results from animal, cell, or clinical studies”. Anyway, as pointed out in my answer to the previous point 4, the Tables directly connected with EA activity on AD are three (Table 12, 13 and 14, revised manuscript) and not exclusively Table 13 (revised manuscript). Concerning other Tables, since the organization of the Review comprehends several themes connected to the central one (as described at the end of the Introduction), to give reader a picture as complete and updated as possible on such vast topic, we believe that Tables are the best reader-friendly tool to summarize information. In addition, given that up to 3 Reviewers have greatly appreciated their presence, we kindly ask the Reviewer not to force us to significantly reduce their number, so as not to displease the other 3 Reviewers. Anyway, to partially satisfy the Reviewer, two Tables (Table 7 and 11, original manuscript) have been moved in Appendix B, where they appear as Table B1 and B2.

  1. Structure and Theme Misalignment: Of the eight sections in the paper, only Section 6 aligns with the stated theme, and even then, it addresses AD treatment mechanisms rather than the theme of "Reducing Oxidative Stress and Inflammation." The lack of focus is evident, as only 1/8 of the content is tangentially related to the theme.

The theme of "Reducing Oxidative Stress and Inflammation" as the main mechanism by which polyphenols and specifically EA ameliorate the conditions of neurodegenerative diseases, including AD, is extensively disserted in Section 5.1. inside Section 5. Concerning Sections 2 and 3, they regard AD which is a topic of this Review. Section 4 regards the potentialities of both ETs, EA and UROs as beneficial antioxidant molecules and explain the reasons because EA is the most researched compound, and it has been selected for this Review. Finally, Section 6 is more in deep involved in the stated theme and in reporting the findings from the in vivo and in vitro experiments carried out to assess EA effects on AD, using different cells and animal models. In our opinion, all sections (which are 7 and not 8 for our distraction, now corrected) are directly related to the theme.

Reviewer 4 Report

Comments and Suggestions for Authors

The text below contains comments on manuscript entitled “Ellagic acid (EA): A Green Multi-Target Weapon Reducing Oxidative Stress and Inflammation Thus Preventing and Ameliorating Alzheimer Disease (AD) Condition”

I have several major concerns and minor comment that I have listed below. To my opinion the manuscript is too long. A clear and specific aim is missing. The authors have tried to present too much figure and tables many of which not necessary or not applicable here. That is why at many places the authors make large deviations from the topic of the title related to AD, for example the introduction of the tables 12 and 13. They are more suitable if the manuscript is on a botanical topic. Another major concern is the 55% of similarities.

I highly advise the authors to reconsider again the whole manuscript and really stay focused on what is the aim of their manuscript. Otherwise the reader is lost and not possible to follow your thoughts.

I believe the abbreviation EA and AD are not necessary in the title. The abbreviations of ellagic acid and Alzheimer Disease can be introduced in the main text.

To my opinion, the introduction is too long, probably some of the text might be separated as an additional section. However, the main drawback of the introduction is the clear and well formed aim of the manuscript. What is the main purpose of the manuscript, what is your main hypothesis, what important issues this manuscript will solve, what gaps in the field will fill and what are your main expectations and outcomes.

I think that figure 7 is not necessary. The meaning of this figure can be presented in 1 or 2 sentences in the main text. I have the same comment for figure 8, 9, 11 and 12.

I am really doubtful if table 7 is necessary.

Scheme 1 needs to be presented with a better quality and also needs to be the author original work.

What are your main conclusions from table 8?

I think that table 11 is also unnecessary. I believe it is not necessary to present the mechanisms of antioxidant activity, the manuscript starts to look more like a textbook for students. Nevertheless, in this section you have already started to focus on ellagic acid and in table 11 you have started to include again quite large number of other antioxidant molecules.

Do you pretend that table 12 is complete and that it summarizes all plant species containing ETs and EA? Is that really the focus of your manuscript? I found this table not necessary. I have the same comment for figure 13.

Author Response

The text below contains comments on manuscript entitled “Ellagic acid (EA): A Green Multi-Target Weapon Reducing Oxidative Stress and Inflammation Thus Preventing and Ameliorating Alzheimer Disease (AD) Condition”

I have several major concerns and minor comment that I have listed below. To my opinion the manuscript is too long. A clear and specific aim is missing. The authors have tried to present too much figure and tables many of which not necessary or not applicable here. That is why at many places the authors make large deviations from the topic of the title related to AD, for example the introduction of the tables 12 and 13. They are more suitable if the manuscript is on a botanical topic. Another major concern is the 55% of similarities.

I highly advise the authors to reconsider again the whole manuscript and really stay focused on what is the aim of their manuscript. Otherwise the reader is lost and not possible to follow your thoughts.

We thank the Reviewer for his/her comments that enable us to explain better the rational and organization of this study and to apply the due modifications to improve it. Since the topic is very vast, the manuscript has resulted naturally long. Anyway, we consider all parts necessary, also to differentiate this work by others already existing in literature and focusing on more restricted fields, which are instead intercorrelated one to each other. The clear and specific aim of this study and a summary of its parts were already reported at the end of Introduction, in the original version of the manuscript. Respecting these aims, the work has been organized as appeared. Anyway, for more clarity and on other suggestions of the Reviewer this part has been modified and divided into the two new separate Sections 1.3. and 1.4. Please see at lines 203-222.

Concerning those parts that Reviewer calls “large deviations from the topic of the title related to AD”, they are instead strictly correlated to EA and AD and are essential to provide readers an all-around scenario and the current state of the art concerning AD and the potentialities of polyphenols and more specifically EA, to prevent and/or treat neurodegenerative diseases including AD. Concerning the presence of too many Figures and Tables, many of which considered not necessary or not applicable here by the Reviewer, we underline again that, since the organization of the Review comprehends several themes connected to the central one (as described at the end of the Introduction), to give reader a picture as complete and updated as possible on such vast topic, we believed that Tables and Figures are the best reader-friendly tool to summarize information. In addition, given that up to 3 Reviewers have greatly appreciated their presence, we kindly ask the Reviewer not to force us to significantly reduce their number, so as not to displease the other 3 Reviewers. As for Table 12 and 13, we consider them appropriate for this Review since EA is a botanical topic.

Although the Editorial Office of IJMS has not still required a revision to reduce duplicated, several parts of the manuscript have been modified to reduce similarities, as asked by the Reviewer.

I believe the abbreviation EA and AD are not necessary in the title. The abbreviations of ellagic acid and Alzheimer Disease can be introduced in the main text.

As asked, abbreviations have been removed by the title.

To my opinion, the introduction is too long, probably some of the text might be separated as an additional section. However, the main drawback of the introduction is the clear and well formed aim of the manuscript. What is the main purpose of the manuscript, what is your main hypothesis, what important issues this manuscript will solve, what gaps in the field will fill and what are your main expectations and outcomes.

On suggestion of the Reviewer, the Introduction, which has been enriched with additional parts on request of other Reviewers, has been fragmented in more subsections (1.1-1.4), one addressing the requests of the Reviewer. Please, consider the structure of the modified Introduction and particularly lines 203-213.

I think that figure 7 is not necessary. The meaning of this figure can be presented in 1 or 2 sentences in the main text. I have the same comment for figure 8, 9, 11 and 12.

We thank the Reviewer for the suggestion. Figures 7, 8, 9, 11 and 12 have been removed by the main text and have been moved in the new Appendix B, before the reference list, where now they appear as Figure B1, B2, B3, B4 and B5.

I am really doubtful if table 7 is necessary.

Table 7 has been moved to Appendix B.

Scheme 1 needs to be presented with a better quality and also needs to be the author original work.

We assure the Reviewer that Scheme 1 is of our production. We made it using ChemDraw Ultra 7.0 and it was published by us for the first time on Antioxidants, 2020 (Ref.17). Anyway, a specification on this fact and the related reference have been included in the Scheme 1 caption (lines 584-585). The quality of Scheme has been improved.

What are your main conclusions from table 8?

Our main conclusions from Table 8 (Table 7 in the revised manuscript) were already inserted under the same Table. Anyway, some other details have been inserted.

I think that table 11 is also unnecessary. I believe it is not necessary to present the mechanisms of antioxidant activity, the manuscript starts to look more like a textbook for students. Nevertheless, in this section you have already started to focus on ellagic acid and in table 11 you have started to include again quite large number of other antioxidant molecules.

We agree with the Reviewer. Consequently, Table 11 has been firstly modified and then moved to Appendix B, as Table B2.

Do you pretend that table 12 is complete and that it summarizes all plant species containing ETs and EA? Is that really the focus of your manuscript? I found this table not necessary. I have the same comment for figure 13.

We do not pretend that the original Table 12 (Table 10 in the revised manuscript) is complete, but our scope was to give a list of the main plants containing ETs and EA in appreciable concentrations, which upon assumption by humans or animal demonstrated beneficial medical effects. Now, this specification is included in the title of Table 10 (revised manuscript). We think that knowing the plants species containing significant concentrations of ETs and EA, could be of great help for organizing a nutrition regime which could be supportive in preventing neurodegenerative diseases as AD. We kindly ask the Reviewer to allow us to maintain Table 10. Figure 13 does not exist in our manuscript.

Reviewer 5 Report

Comments and Suggestions for Authors

The present work is a comprehensive, critical and analytical review on the potential of ellagic acid as a green option to treat human degenerative diseases such as Alzheimer disease. The review follows deductive approach starting with section dealing with the nature of neurodegenerative diseases, through the different approaches to treat them and opens several sections dedicated to the role of ellagic acid and ellagitannins at all as bioactive compounds.

I can suggest the following improvements:

- please check if the references are in accordance with the journal's recommendations;

- please, revise your manuscript for typos and spelling mistakes;

Sincerely!

Author Response

The present work is a comprehensive, critical and analytical review on the potential of ellagic acid as a green option to treat human degenerative diseases such as Alzheimer disease. The review follows deductive approach starting with section dealing with the nature of neurodegenerative diseases, through the different approaches to treat them and opens several sections dedicated to the role of ellagic acid and ellagitannins at all as bioactive compounds.

I can suggest the following improvements:

- please check if the references are in accordance with the journal's recommendations;

- please, revise your manuscript for typos and spelling mistakes;

Sincerely!

As asked, although by using the Mendeley software to insert references they are automatically inserted according to the journal requirements, we have furtherly double-checked them. Additionally, all manuscript has been revised to reduce typos and spelling mistakes.

Round 2

Reviewer 1 Report

Comments and Suggestions for Authors

The authors have answered my questions well and made the necessary changes to the manuscript (ID: ijms-3400708). It looks ready for publication as far as I can tell. Then, I think that the manuscript can be accepted for publication in International Journal of Molecular Sciences.

Author Response

The authors have answered my questions well and made the necessary changes to the manuscript (ID: ijms-3400708). It looks ready for publication as far as I can tell. Then, I think that the manuscript can be accepted for publication in International Journal of Molecular Sciences.

We thank a lot the Reviewer for his/her helpful suggestions and for this positive decision.

Reviewer 3 Report

Comments and Suggestions for Authors

Although the authors have made some revisions, the content still does not fully align with the theme and lacks sufficient focus.

1. The authors stated in their response that “The theme of ‘Reducing Oxidative Stress and Inflammation’ as the main mechanism by which polyphenols, and specifically EA, ameliorate the conditions of neurodegenerative diseases, including AD, is extensively disserted in Section 5.1 within Section 5.” While it is correct that Section 5.1 discusses EA’s role in reducing oxidative stress and inflammation, this discussion does not directly address whether EA exhibits these antioxidant effects in the context of AD treatment, particularly in animal or cell models.

The authors have explained the antioxidant mechanism of EA in isolation, which only demonstrates that EA has potential antioxidant properties. However, this does not necessarily mean that the same mechanism is active during the treatment of AD. When evaluating a treatment for a disease, the first step is to establish its efficacy in addressing the specific condition. Only then can its underlying mechanisms be explored. Many drugs exhibit antioxidant effects, but this does not imply that they rely on this mechanism when treating AD. To claim that EA exerts its antioxidant effects in the context of AD, its efficacy in treating AD must first be demonstrated.

 A compound may exhibit antioxidant properties but still be effective only for other diseases, such as gastrointestinal conditions, rather than AD. In such cases, it cannot be concluded that the compound treats AD via an antioxidant mechanism. Therefore, the discussion in Section 5.1 lacks direct evidence to confirm EA’s antioxidant mechanism specifically in AD treatment. The authors are encouraged to provide experimental or research data directly supporting the role of EA’s antioxidant effects in AD treatment.

2. Regarding Section 4, the authors stated, “Section 4 regards the potentialities of both ETs, EA, and UROs as beneficial antioxidant molecules and explains the reasons why EA is the most researched compound, and it has been selected for this Review.” This section is disproportionately lengthy given the focus of the paper. The title of the manuscript, “Ellagic acid (EA): A Green Multi-Target Weapon that Reduces Oxidative Stress and Inflammation to Prevent and Improve the Condition of Alzheimer’s Disease,” suggests that the structure should center around how EA reduces oxidative stress and inflammation to prevent and treat Alzheimer’s disease, rather than dedicating significant content to explaining why EA was chosen over ETs.

If the primary focus is to justify the selection of EA instead of ETs, then the title should be adjusted to reflect the comparative relationship between these compounds in the context of AD treatment. Otherwise, it is recommended to streamline this section and reorient the content to align with the stated theme of EA’s specific mechanisms and effects on Alzheimer’s disease.

3. Regarding the authors' response, “Concerning Sections 2 and 3, they regard AD, which is a topic of this Review,” the authors provide extensive discussion on current therapeutic approaches and the pathogenesis of AD. While this background information is valuable for setting the stage, the content is disproportionately focused on AD broadly, with relatively little attention given to the core theme of the paper: EA’s role in AD treatment.

Readers are likely drawn to this article for its unique focus on how EA treats AD through antioxidant mechanisms, and this should be the primary content of the manuscript. If readers seek a broader understanding of AD pathogenesis, they might turn to other, more specialized reviews on this topic that offer greater depth. Unfortunately, this paper devotes less than half of its content to its stated theme, which dilutes its uniqueness and focus. It is recommended that the authors significantly reorient the manuscript to emphasize EA’s specific mechanisms and effects in the context of AD treatment, reducing the background content on AD to a concise and targeted introduction.

Author Response

Although the authors have made some revisions, the content still does not fully align with the theme and lacks sufficient focus.

Now we have cleared the idea that the Reviewer has decided to assume and maintain a hostile and almost non-objective attitude towards our manuscript. We kindly make note the Reviewer, that his/her position is not shared by 4 other Reviewers who, after the first revision, share fully positive opinions and would have approved its acceptance. However, for scientific honesty and professional correctness, I am happy to accept and, if possible, resolve the Reviewer's further criticisms.

  1. The authors stated in their response that “The theme of ‘Reducing Oxidative Stress and Inflammation’ as the main mechanism by which polyphenols, and specifically EA, ameliorate the conditions of neurodegenerative diseases, including AD, is extensively disserted in Section 5.1 within Section 5.” While it is correct that Section 5.1 discusses EA’s role in reducing oxidative stress and inflammation, this discussion does not directly address whether EA exhibits these antioxidant effects in the context of AD treatment, particularly in animal or cell models.

The authors have explained the antioxidant mechanism of EA in isolation, which only demonstrates that EA has potential antioxidant properties. However, this does not necessarily mean that the same mechanism is active during the treatment of AD. When evaluating a treatment for a disease, the first step is to establish its efficacy in addressing the specific condition. Only then can its underlying mechanisms be explored. Many drugs exhibit antioxidant effects, but this does not imply that they rely on this mechanism when treating AD. To claim that EA exerts its antioxidant effects in the context of AD, its efficacy in treating AD must first be demonstrated.

 A compound may exhibit antioxidant properties but still be effective only for other diseases, such as gastrointestinal conditions, rather than AD. In such cases, it cannot be concluded that the compound treats AD via an antioxidant mechanism. Therefore, the discussion in Section 5.1 lacks direct evidence to confirm EA’s antioxidant mechanism specifically in AD treatment. The authors are encouraged to provide experimental or research data directly supporting the role of EA’s antioxidant effects in AD treatment.

Dear Reviewer, thank you for this extensive explanation, regarding the necessity of demonstrating an actual correlation between the general antioxidant properties of EA and the effective capacity of such properties to ameliorate AD condition. Anyway, following our design concerning this review, as clearly defined in the Abstract and Introduction (lines 206-214), these demonstrations would have been inserted in the subsequent Section 6, where the results from in vitro and in vivo experiments on AD models, were reported. In section 5.1., we limited ourselves to giving indications on the mechanisms by which EA exert antioxidant powers and behaves as a powerful radical scavenger. What is requested by the Reviewer is available specifically in Table 12, 13 and 14, where several times it is reported that, amelioration in AD conditions after EA administration, was due to the reduction of ROS, anion superoxide etc and other markers of inflammation, as well as to the concomitant increasing of antioxidant enzymes (SOD, CAT, GSH etc).

  1. Regarding Section 4, the authors stated, “Section 4 regards the potentialities of both ETs, EA, and UROs as beneficial antioxidant molecules and explains the reasons why EA is the most researched compound, and it has been selected for this Review.” This section is disproportionately lengthy given the focus of the paper. The title of the manuscript, “Ellagic acid (EA): A Green Multi-Target Weapon that Reduces Oxidative Stress and Inflammation to Prevent and Improve the Condition of Alzheimer’s Disease,” suggests that the structure should center around how EA reduces oxidative stress and inflammation to prevent and treat Alzheimer’s disease, rather than dedicating significant content to explaining why EA was chosen over ETs.

If the primary focus is to justify the selection of EA instead of ETs, then the title should be adjusted to reflect the comparative relationship between these compounds in the context of AD treatment. Otherwise, it is recommended to streamline this section and reorient the content to align with the stated theme of EA’s specific mechanisms and effects on Alzheimer’s disease.

Dear Reviewer, we make kindly note you, that the request of better evidencing the reasons for which EA is preferred to ETs and UROs was one of your comments. Anyway, we make kindly him/her know, that in the first round-revision, we had to satisfy the requests of as many as 5 Reviewers. At least two of them (in addition to this Reviewer) asked me to explain better and in detail why, between ETs, UROs and EA, EA is the compound considered the most promising and therefore the most studied as a possible molecule to be developed and applied clinically, to combat various neurodegenerative diseases including AD, and so, was selected for this Review. The disquisition of this facts, that the Reviewer considers too long, is due to these requests, including his/her one.

  1. Regarding the authors' response, “Concerning Sections 2 and 3, they regard AD, which is a topic of this Review,” the authors provide extensive discussion on current therapeutic approaches and the pathogenesis of AD. While this background information is valuable for setting the stage, the content is disproportionately focused on AD broadly, with relatively little attention given to the core theme of the paper: EA’s role in AD treatment.

Readers are likely drawn to this article for its unique focus on how EA treats AD through antioxidant mechanisms, and this should be the primary content of the manuscript. If readers seek a broader understanding of AD pathogenesis, they might turn to other, more specialized reviews on this topic that offer greater depth. Unfortunately, this paper devotes less than half of its content to its stated theme, which dilutes its uniqueness and focus. It is recommended that the authors significantly reorient the manuscript to emphasize EA’s specific mechanisms and effects in the context of AD treatment, reducing the background content on AD to a concise and targeted introduction.

Dear Reviewer, before to emphasize EA’s specific mechanisms and effects in the context of AD treatment, our aim was first to give readers an all-round vision of both the considered disease (AD), its possible causes, the available therapeutic options and those innovative under clinical trials, to arrive at the alternative natural therapeutic opportunities, which include mainly the antioxidant polyphenols to be used as ameliorative molecules. In our opinion, this part is essential so that readers can have a clear understanding of the state of the art of the topic, in which EA will act as main player, without having to resort to consulting medical manuals, that may be more difficult to understand for non-experts. An extensive dissertation on ETs, EA and UROs which are all molecules metabolically connected, and then specifically on EA in relation to its involvement in the possible treatment of AD, constitute the core of this study. Results from the in vitro and in vivo studies developed so far on AD treatment by EA administration to AD models, conclude this Review. We are confident that readers of IJMS, as the other 4 Reviewers, will appreciate our work and its organization.